# Limited Therapeutic Options in Mexico for the Treatment of Urinary Tract Infections

**DOI:** 10.3390/antibiotics11111656

**Published:** 2022-11-18

**Authors:** Guadalupe Miranda-Novales, Karen Flores-Moreno, Yolanda López-Vidal, Samuel Ponce de León-Rosales

**Affiliations:** 1Analysis and Synthesis of Evidence Research Unit, Mexican Institute of Social Security, Mexico City 06720, Mexico; 2Microbiome Laboratory, Faculty of Medicine, National Autonomous University of Mexico, Mexico City 04360, Mexico; 3Microbiology and Parasitology Department, Faculty of Medicine, National Autonomous University of Mexico, Mexico City 04360, Mexico; 4University Program for Health Research, National Autonomous University of Mexico, Mexico City 04510, Mexico

**Keywords:** urinary tract infection, uropathogen, antimicrobial resistance, *Escherichia coli*, *Klebsiella pneumoniae*

## Abstract

The rise in antimicrobial resistance (AMR) has complicated the management of urinary tract infections (UTIs). The objective of this study was to evaluate the antimicrobial susceptibility patterns of *Escherichia coli* and *Klebsiella pneumoniae.* Design: prospective observational study. Bacteria were classified as susceptible or resistant to ampicillin-sulbactam, amikacin, gentamicin, ciprofloxacin, norfloxacin, nitrofurantoin, trimethoprim-sulfamethoxazole (TMP/SMZ), ertapenem, meropenem, and fosfomycin. The sensitivity to fosfomycin and chloramphenicol was evaluated by the disk diffusion method. Statistical analysis: the chi-square test and Fisher’s exact test were used to compare differences between categories. A *p* value < 0.05 was considered statistically significant. Isolates were collected from January 2019 to November 2020 from 21 hospitals and laboratories. A total of 238 isolates were received: a total of 156 *E. coli* isolates and 82 *K. pneumoniae* isolates. The majority were community-acquired infections (64.1%). Resistance was >20% for beta-lactams, aminoglycosides, fluoroquinolones, and TMP/SMZ. For *E. coli* isolates, resistance was <20% for amikacin, fosfomycin, and nitrofurantoin; for *K. pneumoniae,* amikacin, fosfomycin, chloramphenicol, and norfloxacin. All were susceptible to carbapenems. *K. pneumoniae* isolates registered a higher proportion of extensively drug-resistant bacteria in comparison with *E. coli* (*p* = 0.0004). In total, multidrug-resistant bacteria represented 61% of all isolates. Isolates demonstrated high resistance to beta-lactams, fluoro-quinolones, and TMP/SMZ.

## 1. Introduction

Urinary tract infections (UTIs) are one of the most common reasons for consultation that involves the prescription of an antibiotic. Several guidelines allow the use of empirical antibiotics without the need for a urine culture and a susceptibility report on the causal agent. For many years, this was a common and accepted practice at the first level of care, mostly in cases of acute uncomplicated cystitis. The notable rise in antimicrobial resistance (AMR) has complicated the management and efficacy of empirical antibiotics and constitutes a real challenge [1]. It is believed that UTIs represent a multi-positional problem, as they result in a decrease in quality of life (QoL), especially in cases of recurrence, complications, and sequelae; they also consume a large amount of economic and human resources [2].

AMR resistance changes markedly in different geographical areas, and unfortunately, there are no periodic reports from all countries. A systematic review of AMR in uropathogens from the Asia-Pacific region found a prevalence of resistance against trimethoprim/sulfamethoxazole (TMP/SMZ), ciprofloxacin, and ceftriaxone from 33 to 90%, against nitrofurantoin from 2.7 to 31.4%, and against fosfomycin at 1.8% [3]. In contrast, a review from Iran showed that one of the most effective antibiotics for *Escherichia coli* was ciprofloxacin, along with nitrofurantoin, imipenem, and chloramphenicol [4]. One factor to consider in selecting an empirical antibiotic is the regional prevalence of antimicrobial multidrug-resistant (MDR) bacteria. Some reports have found this characteristic with a higher frequency in *K. pneumoniae* isolates (40.4%) in comparison with *E. coli* (23.3%) [5].

In 2017, the Universidad Nacional Autónoma de México (UNAM) proposed an action plan to control antimicrobial resistance in Mexico through the University Program for Health Research (*Programa Universitario de Investigación en Salud*, PUIS). One of the first tasks of this plan was to invite health personnel from institutions and laboratories to send information, thus establishing the status of antimicrobial resistance in Mexico (the PUCRA network is named after its acronym in Spanish: *Plan Universitario de Control de la Resistencia Antimicrobiana*). In the first report, information from 12,151 isolates from urine cultures collected during 2016 and 2017 (90% corresponded to *Escherichia coli* and 10% to *Klebsiella pneumoniae*) found that in *E. coli,* median resistance to amikacin, imipenem, meropenem, and nitrofurantoin was <10%; *K. pneumoniae* isolates also showed low resistance to amikacin and carbapenems, but median resistance to nitrofurantoin was 52%. For both enteric bacteria, median resistance to cephalosporins, ciprofloxacin, and TMP/SMZ was >40% [6]. Fosfomycin and chloramphenicol susceptibilities were not available. Another network in Mexico (Network for Research and Surveillance of Drug Resistance [*Red Temática de Investigación y Vigilancia de la Farmacorresistencia* (INVIFAR)]) reported high resistance (>40%) in *E. coli* since 2009 [7].

AMR resistance in uropathogens is recognized as a serious problem in Latin American countries. In an analysis of data obtained from the Global Burden of Diseases, Injuries, and Risk Factors Study, *Escherichia coli* and *Klebsiella pneumoniae* accounted for over 50% of the deaths attributable to and associated with AMR [8]. 

Before 2010, in Mexico, antibiotics were sold over the counter and AMR in common pathogens was in the top places in comparison with other Latin American countries. Some regulatory changes have been introduced. Today, a doctor’s prescription is needed to purchase an antibiotic at a drug store. There is no proper evaluation of the impact of the regulatory measure, but most of the local published information is consistent with an increase in AMR [9,10,11,12,13].

Fosfomycin is an old drug but an interesting candidate for the treatment of infections due to MDR bacteria [14,15]. Chloramphenicol, an old broad-spectrum antibiotic, has been included to understand the genomic diversity and genotypic presence of antimicrobial resistance in MDR *E. coli* and *K. pneumoniae* isolates by whole genome sequencing (WGS) [16].

The objective of this study was to evaluate the antimicrobial susceptibility patterns of *Escherichia coli* and *Klebsiella pneumoniae* isolates causing UTIs and to test in vitro antimicrobial susceptibility to fosfomycin and chloramphenicol.

## 2. Results

### 2.1. Origin of Isolates

#### Isolates Were Collected from January 2019 to November 2020

Twenty-one hospitals and laboratories from eight states of the Mexican Republic (12 located in Mexico City, two in Guanajuato, two in Puebla, and one each in Mexico State, Durango, Jalisco, Nuevo León, and Morelos) sent a minimum of ten urine culture isolates. Characteristics of hospitals/institutions: seven specialty hospitals, five general hospitals, three pediatric hospitals, three private reference laboratories, two tertiary-care centers, and one private hospital (Table 1).

In total, 238 isolates were received: 156 *Escherichia coli* isolates and 82 *Klebsiella pneumoniae* isolates. Most of the isolates corresponded to community-acquired infections (64.1%) from non-hospitalized (57.2%) and adult patients (>18 years, 87%). *Escherichia coli* isolates were more common in females (72%), with a similar sex distribution for *Klebsiella pneumoniae* isolates (52% in females and 48% in males).

### 2.2. Antimicrobial Susceptibility Patterns

Resistance was high for most of the tested antibiotics, including beta-lactams, aminoglycosides, fluoroquinolones, and TMP/SMZ. For *E. coli* isolates, resistance was less than 20% only for amikacin (0.6%), fosfomycin (9.5%), and nitrofurantoin (8%). For *K. pneumoniae* isolates, the antibiotics with the lowest resistance were amikacin (1%), fosfomycin (10%), chloramphenicol (16%), and norfloxacin (19%) (Figure 1). All isolates were susceptible to ertapenem and meropenem.

#### Extended Spectrum Beta-Lactamase Producers and MDR Bacteria

There was no difference in the frequency of ESBL producers between *E. coli* and *K. pneumoniae* isolates (56% vs. 52%, respectively, *p* = 0.62). *E. coli* ESBL-producers were more resistant than ESBL-negative isolates with a significant statistical difference (*p* < 0.01) for the following antibiotics: ampicillin/sulbactam, gentamicin, ciprofloxacin, and norfloxacin. For *K. pneumoniae* isolates, differences were greater for ampicillin/sulbactam, gentamicin, ciprofloxacin, nitrofurantoin, and TMP/SMZ. For two of them (gentamicin and TMP/SMZ), resistance dropped to < 20% (Table 2).

In agreement with these results, *Klebsiella pneumoniae* isolates registered a higher proportion of XDR bacteria compared to *E. coli* (*p* = 0.0004), although the number of isolates was low. However, for the *E. coli* isolates, the proportion of MDR bacteria showed a statistically significant difference in comparison with *K. pneumoniae* (68% vs. 48%). Overall, MDR bacteria represented the most frequent characteristic of all isolates (61%). There was no difference for isolates resistance to at least one antibiotic in two antimicrobial categories (*p* = 0.22) (Table 3).

A small number of isolates (*n* = 31) corresponded to pediatric patients. Seventy-four percent of them were *E. coli.* In comparison with isolates from adult patients, susceptibility patterns were similar (Table 4). A major problem was observed in *K. pneumoniae*, 7/8 isolates were ESBL producers and showed high resistance rates. Fosfomycin was the only active oral agent.

## 3. Discussion

To treat UTIs effectively, the correct antibiotic should be used at the correct dose for the shortest effective duration of therapy possible [17,18]. Regrettably, in some countries such as Mexico, clinical practice guidelines are not updated [19,20] and first-line empirical treatments still include TMP/SMZ, cephalexin, and amoxicillin, despite the availability of local information in several publications [6,7,10,11,12,13,21].

One of the consequences of the lack of adequate and prompt treatment is the progression of infection, leading to the development of a potentially fatal systemic infection, especially in high-risk patients. In the Global Prevalence of Infections in Urology (GPIU) study, the antibiotics most prescribed were fluoroquinolones (35%), cephalosporins (27%), and penicillin (16%). The resistance rates of all antibiotics tested against the isolates other than carbapenems were higher than 10% [22].

The results of this study validate the existence of the AMR problem in uropathogens. Most of the published information on ESBL-positive Enterobacteriaceae is from bloodstream and intraabdominal infections. In general, these isolates are also resistant to several antibiotics, such as fluoroquinolones and TMP/SMZ, which are no longer suitable for the empiric treatment of UTI [23]. In this study, which included isolates from outpatients, inpatients, and reference laboratories, demonstrated that more than half of the isolates were ESBL producers, both in community and healthcare-associated infections. Thus, oral options for the treatment of uncomplicated UTIs in Mexico are limited to nitrofurantoin and fosfomycin for *E. coli*, and only fosfomycin for *K. pneumoniae* isolates. Almost 70% of *E.coli* isolates were MDR, leaving amikacin and carbapenems as the only options for acute pyelonephritis and complicated cases.

Even in this reduced sample of pediatric patients, high resistance rates and MDR were the common features in most of the isolates, in contrast with other reports, with resistance rates to first line treatments between 10–40% [24]. We need to consider the referral bias in our population, as all pediatric patients are seen at a tertiary-care level hospital. Well-known risk factors such as pre-existing conditions, previous antibiotic treatments, and hospitalization contribute to the selection of resistant strains [25]. As shown in one study, discordant empirical UTI treatment in hospitalized children can be effective in 50% of the cases, but patients with risk factors need a different approach [26]. Half of the patients with a therapeutic failure to an empirical treatment is not acceptable. Before the antibiotic is prescribed, underlying conditions, drug interactions, recurrence of events, and renal function need to be evaluated in each case.

The crisis regarding available options will be worse for institutions with limited drugs (e.g., fosfomycin is not available in the Mexican Institute of Social Security). Over 50% of the isolates are MDR, so in some cases, the only alternative will be carbapenems. The increased use of this group of antibiotics is responsible for the collateral damage and the emergence of the carbapenem-resistant pathogens (enteric bacteria, *Acinetobacter baumannii*, and *Pseudomonas* sp.) Although chloramphenicol showed good in vitro activity, most of a chloramphenicol dose is metabolized by the liver into inactive products, and only 5–15% of chloramphenicol is excreted unchanged in the urine. Therefore, it is not a useful alternative for UTIs.

This study has several limitations: the clinical information is minimal, risk factors, characteristics of infection, pre-existing conditions, recent hospitalization, previous use of antibiotics, response to treatment, and outcome are not available. However, our results are consistent with several reports in the country and worldwide. There is an urgent need to implement antimicrobial stewardship programs targeting UTIs. Although most studies focus on adult patients, the principles are the same for all age groups. The five-D model includes correct diagnosis, the right drug, dose, and duration, and finally de-escalation (narrow spectrum antibiotics, or stopping antibiotics based on culture results) [27]. Anti- microbial stewardship programs have proven their effectiveness in urgent care [28] as well as in hospitalized patients with potentially fatal outcomes [29].

Since new effective antibiotics are not on the horizon, different alternatives to antibiotics are being evaluated: vaccines, small compounds targeting adhesion, urease, bacterial capsules, nutraceuticals, bacteriophages, and probiotics. Nevertheless, further research is needed before definitive recommendations can be issued [30,31].

## 4. Materials and Methods

Design: prospective observational. Institutions and laboratories in the PUCRA network were asked to send non-duplicate *Escherichia coli* and *Klebsiella pneumoniae* isolates. Inclusion criteria: patients of all ages and both sexes, diagnosed with a symptomatic UTI, regardless of acquisition (community or healthcare-associated infection), history of urologic procedures, previous infections, or prior antibiotic use. Only one non-duplicate isolate per patient was included. Exclusion criteria: asymptomatic bacteriuria.

Bacterial identification and determination of the antimicrobial susceptibility of each isolate were performed with standard microbiological techniques. Microbial identification was done with the GN ID card and testing of susceptibility to antimicrobial drugs was performed using the microdilution method (VITEK 2 XL^®^ BioMérieux) with AST-271 and AST-272 cards. Bacteria were classified as producers of extended spectrum beta-lactamases (ESBL) and susceptible or resistant to ampicillin-sulbactam, amikacin, gentamicin, ciprofloxacin, norfloxacin, nitrofurantoin, TMP/SMZ, ertapenem, meropenem, and fosfomycin, according to the CLSI (Clinical and Laboratory Standards Institute) guidelines [32].

Sensitivity to fosfomycin and chloramphenicol was evaluated by the disk diffusion method, with fosfomycin 200 µg/50 µg of glucose 6-phosphate and chloramphenicol 30 µg disks (Oxoid TM). The Mueller–Hinton agar plates (BD TM) were incubated for 24 h at 35 °C and the susceptibility results for fosfomycin were interpreted according to the CLSI M100 [32] breakpoints (zone diameter, mm) for *E. coli* extrapolated to enteric bacteria (≥16 susceptible, 13–15 intermediate, and ≤12 resistant). Breakpoints for chloramphenicol are ≥18 susceptible, 13–17 intermediate, and ≤12 resistant. *Escherichia coli* ATCC^®^ 25,922 was used as a quality control strain. 

Definitions: MDR was defined as acquired non-susceptibility to at least one agent in three or more antimicrobial categories; XDR (extensively drug-resistant) was defined as non-susceptibility to at least one agent in all but two or fewer antimicrobial categories (i.e., bacterial isolates remain susceptible to only one or two categories); and PDR (pandrug-resistant) was defined as non-susceptibility to all agents in all antimicrobial categories [33].

Statistical analysis: Categorical data are presented as frequencies and percentages. The chi-square test and Fisher’s exact test were used to compare differences between categories. A *p* value < 0.05 was considered as statistically significant.

The study was conducted in accordance with the Declaration of Helsinki and approved by the Ethics and Research Commissions of the Faculty of Medicine and registered with the number: No. FM-DI/060/2019 on 5 January 2019. Patient consent was deemed unnecessary by the ethics and research commissions due to the design of the study and because data on isolates were not linked to an identifiable person.

## 5. Conclusions

*Escherichia coli* isolates demonstrated high resistance to beta-lactams, fluoroquinolones, and TMP/SMZ. For oral treatment in Mexico, only nitrofurantoin and fosfomycin remain as effective alternatives. For *Klebsiella pneumoniae*, the only active oral antibiotic is fosfomycin. Amikacin and carbapenems are the drugs of choice for complicated infections. MDR bacteria were the most frequent characteristic of all isolates (61%).

## Figures and Tables

**Figure 1 antibiotics-11-01656-f001:**
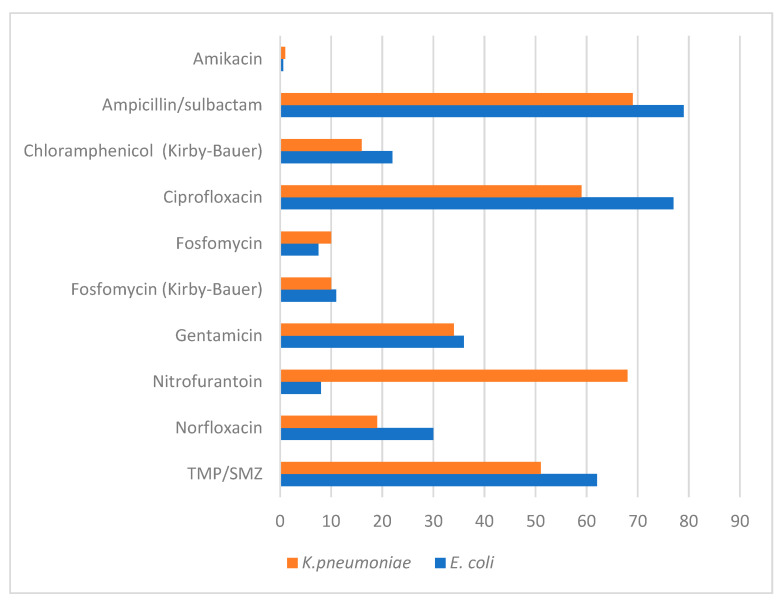
Antibiotic resistance percentages in *Escherichia coli* (*n* = 156) and *Klebsiella pneumoniae* (*n* = 82) isolates from urine cultures.

**Table 1 antibiotics-11-01656-t001:** Characteristics of hospitals/institutions participating in the study.

Name of Hospital or Institution (Location)	Type of Hospital	Number of Beds
1. Instituto Nacional de Neurología y Neurocirugía “Dr. Manuel Velasco Suárez” (Mexico City)	Specialty (neurology and neurosurgery)	126
2. Instituto Nacional de Cancerología (Mexico City)	Specialty (oncology)	188
3. Instituto Nacional de Cardiología “Dr. Ignacio Chávez” (Mexico City)	Specialty (cardiology)	213
4. Hospital General “Dr. Manuel Gea González” (Mexico City)	General	107
5. UMAE Hospital de Cardiología “Dr. Luis Méndez”, Centro Médico Nacional Siglo XXI, IMSS (Mexico City)	Specialty (cardiology)	170
6. Hospital Civil de Guadalajara “Fray Antonio Alcalde” (Jalisco)	General	843
7. Hospital General Regional Nº 200 Tecámac, IMSS (State of Mexico)	General (regional)	249
8. Hospital General de México “Dr. Eduardo Liceaga” (Mexico City)	General	842
9. Hospital General de Durango (Durango)	General	126
10. UMAE 34 Hospital de Cardiología Monterrey, IMSS (Nuevo León)	Specialty (cardiology)	200
11. Hospital Regional de Alta Especialidad del Bajío (Guanajuato)	Tertiary-care (regional)	184
12. Hospital Central Sur de Alta Especialidad de Petróleos Mexicanos (Mexico City)	Tertiary-care	140
13. Laboratorios Ruiz (Puebla)	Private laboratory	Not applicable
14. Instituto Nacional de Rehabilitación “Luis Guillermo Ibarra Ibarra” (Mexico City)	Specialty	228
15. UMAE de Pediatría, Centro Médico Nacional Siglo XXI, IMSS (Mexico City)	Specialty (pediatric)	184
16. Asesores Especializados en Laboratorios, Puebla (AEL)	Private laboratory	Not applicable
17. Hospital Aranda de la Parra (Guanajuato)	Private	109
18. CARPERMOR. Laboratorio de Referencia Internacional (International Reference Laboratory)	Private laboratory	Not applicable
19. Hospital del Niño Morelense (Morelos)	Specialty (pediatric)	38
20. Instituto Nacional de Enfermedades Respiratorias “Dr. Ismael Cosío Villegas” (Mexico City)	Specialty (pulmonology)	175
21. Hospital Pediátrico de la Villa (Mexico City)	General (pediatric)	53

**Table 2 antibiotics-11-01656-t002:** Resistance in *Escherichia coli* and *Klebsiella pneumoniae* isolates according to ESBL * category.

	*Escherichia coli* *n* = 156			*Klebsiella pneumoniae* *n* = 82		
	ESBL+	ESBL-		ESBL+	ESBL-	*p* Value **
Total	56% (87)	44% (69)		52% (43)	48% (39)	0.62
Antibiotic	% Resistant isolates (*n*)		*p* value	% Resistant isolates (*n*)		*p* value
Ampicillin/ Sulbactam	92 (80)	62 (43)	0.00003	100 (43)	35.9 (14)	<0.00001
Amikacin	1 (1)	0 (0)	NA	2 (1)	0 (0)	NA
Gentamicin	47 (41)	23 (15)	0.001	61 (26)	5 (2)	<0.00001
Ciprofloxacin	95 (83)	54 (37)	<0.00001	88 (38)	26 (10)	<0.00001
Norfloxacin	90 (51)	47 (23)	<0.00001	25 (7)	13 (4)	0.22
Nitrofurantoin	10 (8)	6 (4)	0.31	85(35)	50 (19)	0.0008
TMP/SMZ	63 (53)	61 (40)	0.76	85 (35)	13 (5)	<0.00001
Fosfomycin	14 (12)	7 (5)	0.19	7 (3)	13 (5)	0.3
Chloramphenicol (Kirby–Bauer)	26 (23)	16 (11)	0.11	16 (7)	15 (6)	0.91

* ESBL = extended spectrum beta-lactamase producer. ** Chi-square or Fisher’s exact test if applicable. NA = not applicable.

**Table 3 antibiotics-11-01656-t003:** Classification of uropathogens in multidrug-resistant (MDR) and extensively drug-resistant (XDR) bacteria.

Resistance Characteristic *	*Escherichia coli*		*Klebsiella pneumoniae*		*p* Value
	N	%	N	%	
Two	45	29	30	36	0.22
MDR	106	68	39	48	0.002
XDR	5	3	13	16	0.0004
PDR	0	0	0	0	-
Total (*n*)	156	100	82	100	

* Two: non-susceptible to at least one agent in two antimicrobial categories, MDR: non-susceptible to at least one agent in three or more antimicrobial categories, XDR: non-susceptible to at least one agent in all but two or fewer antimicrobial categories (i.e., bacterial isolates remain susceptible to only one or two categories), and PDR (pandrug-resistant): non-susceptibility to all agents in all antimicrobial categories.

**Table 4 antibiotics-11-01656-t004:** Resistance in *Escherichia coli* and *Klebsiella pneumoniae* isolates from adult and pediatric patients according to ESBL * category.

	*Escherichia coli*				*Klebsiella pneumoniae*			
	Adult (*n* = 133)		Pediatric (*n* = 23)		Adult (*n* = 74)		Pediatric (*n* = 8)	
	ESBL+	ESBL-	ESBL+	ESBL-	ESBL+	ESBL-	ESBL+	ESBL-
Total	73 (55%)	60 (45%)	14 (61%)	9 (39%)	37 (50%)	37 (50%)	7 (87%)	1 (13%)
Antibiotic	Resistance %							
Ampicillin/ sulbactam	90	65	100	44	100	32	86	100
Amikacin	1	0	0	0	0	0	28	0
Gentamicin	48	23	43	0	57	5	86	0
Ciprofloxacin	90	55	93	11	86	24	43	0
Nitrofurantoin	11	7	0	0	84	49	62	100
TMP/SMZ	59	58	93	66	89	16	62	0
Fosfomycin	5	5	28	0	8	13	0	0

* ESBL = extended spectrum beta-lactamase producer.

## Data Availability

Databases supporting the results can be provided upon request.

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
