# Peer review of "Limited Therapeutic Options in Mexico for the Treatment of Urinary Tract Infections"

_antibiotics, 2022, doi:10.3390/antibiotics11111656_

Round 1
Reviewer 1 Report
2.1.1. Isolates Were Collected from January 2019 to November 2020 in the abstract is: Isolates were collected from October 2018 to November 2020 from 21 hospitals and laboratories.
Author Response
Dear reviewer,
Thank you for the observation. We have corrected the inconsistency in dates. Isolates were collected from January 2019 to November 2020
Reviewer 2 Report
This report is an interesting observational study, investigating the emergence of resistant strains as causative agents in urinary tract infections. This is a very important issue in the clinical practice, potentially limiting therapeutic options for these diseases. The topic is interesting and the study is generally well conducted even if with several limitations, as also highlighted by authors.
These are my suggestions:
- The introduction appears too long and hard to read. I suggest shortening it and limiting the exposition of existing reports summarizing only relevant results.
- I think that a reference to different age setting could add a significant scientific improvement to this study, considering numerous recent reports exploring the rise of antibiotic resistance strains in pediatric population. A comparison of results of the adult population with some similar multicentric pediatric reports could also be interesting for the discussion. I suggest considering the following articles regarding this aspect , with a particular focus on high resistance to beta-lactams in E. coli isolates (PMID: 29271736; PMID: 34356576).
- Authors state that the main limitation of this study is the lack of clinical information. Are there any information available about the clinical conditions in which these coltures have been performed (symptomatic vs asymptomatic)? Data about the population included would also be relevant (e.g. pediatric vs adult patients? Comorbidities? previous UTIs?). A referral to therapeutic response, if available, would also be nice for clinicians and I would mention it in the limitations (see as example PMID: 35204849).
- The first paragraph of the discussion is too wordy. I suggest shortening it and adding only a summary of these information in the introduction.
- The issue of emerging resistance is very clinically relevant, and authors reported a surprisingly high rate of resistances. A recently explored strategy in order to limit resistances is represented by antimicrobial stewardship programs. I suggest expanding this topic in the discussion, referring to existing stewardship interventions in UTIs setting or in other infectious disease (see for example PMID: 34431702; PMID: 35956160; PMID: 2987775)
Author Response
Dear Reviewer,
Thank you very much for all your insightful comments. We have addressed all of them point by point in the attached document.

Reviewer 3 Report
Dear author,
Very interesting article, but article would be improve.
Design can be improve and should be include inclusion and exlusion criteria. Ethical approve should be include (number and others important information ) .
Author Response
Dear reviewer,
Thank you for your comments.
Point 1. Response. Inclusion and exclusion criteria were added in section 4. Material and methods.
Inclusion criteria: patients of all ages and both sexes, diagnosed with a symptomatic UTI, regardless of acquisition (community or healthcare-associated infection), history of urologic procedures, previous infections, or prior antibiotic use. Only one non-duplicate isolate per patient was included. Exclusion criteria: asymptomatic bacteriuria.
Point 2. Response
Ethical statement was modified to include the required information.
The study was conducted in accordance with the Declaration of Helsinki and approved by the Ethics and Research Commissions of the Faculty of Medicine and registered with the number: No. FM-DI/060/2019 on January 5th, 2019. Patient consent was deemed unnecessary by the ethics and research commissions due to the design of the study and because data on isolates were not linked to an identifiable person.
Round 2
Reviewer 2 Report
Quality of paper was improved in the present form .